# Significant Changes in Serum MicroRNAs after High Tibial Osteotomy in Medial Compartmental Knee Osteoarthritis: Potential Prognostic Biomarkers

**DOI:** 10.3390/diagnostics11020258

**Published:** 2021-02-07

**Authors:** Yoon Hae Kwak, Dae-Kyung Kwak, Hyun-Soo Moon, Nan Young Kim, Jae-Sung Yee, Je-Hyun Yoo

**Affiliations:** 1Division of Orthopaedic Surgery, Severance Children’s Hospital, Yonsei University College of Medicine, Seoul 03722, Korea; yoonhaekwak@yuhs.ac; 2Department of Orthopaedic Surgery, Hallym University Sacred Heart Hospital, Hallym University College of Medicine, Anyang 14068, Korea; limitkdk@gmail.com (D.-K.K.); oshsdesu@gmail.com (H.-S.M.); id0917@naver.com (J.-S.Y.); 3Hallym Institute of Translational Genomics & Bioinformatics, Hallym University Medical Center, Anyang 14068, Korea; honeyny78@gmail.com

**Keywords:** knee, medial compartmental osteoarthritis, high tibial osteotomy, serum miRNA, prognosis

## Abstract

High tibial osteotomy (HTO) is an effective alternative for medial compartmental knee osteoarthritis (OA). Circulating microRNAs (miRNAs) are known to serve as OA-related biomarkers. The present study investigated the differential expression of serum miRNAs before and after HTO to identify potential miRNAs as prognostic biomarkers. miRNA-polymerase chain reaction (PCR) arrays were used to screen for miRNAs in the serum at preoperative and 6-month postoperative time points from six patients, and the differentially expressed miRNAs identified in the profiling stage were validated using real-time PCR at post-operative months 6 and 18 in 27 other HTO-treated patients. Among 84 miRNAs involved in the inflammatory process, three (miR-19b-3p, miR-29c-3p, and miR-424-5p) showed differential expression patterns in the profiling stage (*p* = 0.011, 0.015, and 0.021, respectively). Levels of these three and four other miRNAs (miR-140-3p, miR-454-3p, miR-let-7e-5p, and miR-885-5p) known to be related to OA progression were evaluated in the serum of 27 patients. Only four miRNAs (miR-19b-3p, miR-140-3p, miR-454-3p, and miR-let-7e-5p) were significantly upregulated at postoperative month 6 (*p* = 0.003, 0.005, 0.004, and 0.004, respectively), and only miR-140-3p was significantly upregulated up to 18 months after operation (*p* = 0.003). Together, this study reveals the significantly upregulated serum miRNAs after HTO as potential prognostic biomarkers; however, further studies are warranted to elucidate their clinical implications.

## 1. Introduction

Osteoarthritis (OA) is degenerative joint disease related to functional disability with age. Progression of injured articular cartilage is an irreversible process, leading to its permanent damage. OA prevalence is increasing in the aging population, and OA treatment is a huge burden on the healthcare system worldwide [1]. In patients with medial compartmental OA with varus alignment, high tibial osteotomy (HTO) is a well-known reliable treatment modality [2,3]. Arthroscopic findings in several clinical studies have revealed repair process of the degenerated cartilage following realignment of the mechanical axis of the lower leg. HTO in medial compartmental knee OA offers excellent intermediate and long-term survival [4,5,6].

The pathogenesis of OA is considered a multifactorial and complex process mediated by a close interaction among genetic, mechanical, and environmental factors. The effects of dysregulation at the gene level during the pathogenic process of OA have recently been documented.

MicroRNAs (miRNAs) are small non-coding RNAs comprising 19–25 nucleotides that regulate gene expression post-transcriptionally through mRNA degradation or translational suppression by binding to the 3′-untranslated region of target genes [7]. The roles of miRNAs in cartilage degeneration have recently been reported [8,9]. Most studies that quantified miRNA expression in OA have focused on cartilage or cell lines from the tissue. However, miRNAs are also detected in circulatory body fluids of patients with other pathologies [10,11]. Some studies have demonstrated the association between levels of some circulating miRNAs and diagnosis/prognosis of several diseases, and highlighted their role as potential biomarkers [9,12,13].

Most studies on HTO in medial compartmental knee OA have focused on clinical outcomes in terms of change in the mechanical load on the knee joint resulting from the realignment of the mechanical axis [14,15,16]. Regeneration of degenerated articular cartilage in the medial compartmental knee OA can be expected following HTO without any additional procedures, although the detailed mechanism is still unclear [17]. Despite several studies related to HTO in medial compartmental knee OA and miRNA expression in OA, little is known about the changes in miRNA expression patterns after HTO in medial compartmental knee OA and their clinical implications.

Our previous study revealed significant alterations in the miRNA expression in the synovial fluid (SF) after HTO in medial compartmental knee OA; this effect may be potentially related to clinical and radiographic improvements and subsequent cartilage regeneration [18]. Dysregulated expression of miRNAs in SF has been related to cartilage degradation during OA progression. Significant changes in SF miRNAs after HTO in medical compartmental knee OA may be important findings to predict OA prognosis after HTO. However, obtaining SF aspirate may be invasive and complicated, and require several cumbersome processes. Therefore, this prospective study was conducted to detect serum miRNAs as potential biomarkers for prognosis after HTO. Identification of circulating serum miRNAs with significantly altered expression patterns after HTO would be a promising strategy to highlight their role as biomarkers to evaluate prognosis and clinical and radiological improvement.

## 2. Materials and Methods

### 2.1. Study Subjects

Patients with medial compartmental knee OA who underwent medial open-wedge HTO between November 2015 and February 2017 were prospectively enrolled in this study. This study was approved by the Institutional Review Board of Hallym University Sacred Heart Hospital (protocol code: 2015-I101, Date: 12 October 2015).

Patients who met the following inclusion criteria were finally enrolled in this study: (1) medial compartmental knee OA confirmed on plain radiographs including Rosenberg and long standing views (2) no ligament injuries and lateral meniscus tear on MRI and intraoperative arthroscopic finding (3) no degenerative diseases diagnosed on other joints and spine. All patients had degenerative tear of medial meniscus accompanied by medial compartmental knee OA, confirmed on MRI and intraoperative arthroscopic finding, although the tear site and pattern were slightly different for each patient.

Six patients were recruited for the profiling stage, all of which were female with a mean age of 57.7 years (52 to 65 years). (Table 1). All patients were Kellgren–Lawrence grade (K-L grade) II or III OA only in the medial compartment of the affected knee.

Validation of the miRNAs profiled was carried out by recruiting 27 patients, including 21 female and 6 male patients with a mean age of 57.4 years (46 to 65 years). All patients were K-L grade II or III OA only in the medial compartment. Serum specimens were collected from each patient prior to HTO and at 6-month follow-up (Figure 1). Serum specimens were additionally collected at an outpatient clinic 18 months after surgery in 20 patients who agreed to participate in an 18-month follow-up study (Table 2).

Clinical assessment for pain and functional improvement was performed using a visual analogue scale (VAS) score and the Western Ontario McMaster Universities Osteoarthritis Index (WOMAC) score, both preoperatively and 6 and 18 months after surgery. Radiological evaluation was performed using standing full-length anteroposterior radiographs of the lower limbs taken preoperatively and 6 months after surgery in all cases. These radiographs were obtained to ensure that the patella was directly facing the front. Radiographic changes after HTO were evaluated by measuring the mechanical axis and weight-bearing line (WBL) ratio, which was defined as the perpendicular distance from the WBL to the medial edge of the tibial plateau divided by the width of the tibial plateau. The mechanical axis was defined as the angle subtended by a line drawn from the centre of the femoral head to the centre of the knee and a line drawn from the centre of the knee to the centre of the talus. The WBL was drawn from the centre of the femoral head to the centre of the superior articular surface of the talus [19].

### 2.2. Sampling Strategy and miRNA Analysis

#### 2.2.1. Sample Collection and RNA Extraction from the Serum

First, 10 mL blood samples were collected from all patients and the serum was separated by centrifugation of blood samples within 4 h of collection to ensure consistent pre-analytical conditions. Centrifugation was performed for 10 min at 1900× *g* and 4 °C, followed by for 10 min a 16,000× *g* and 4 °C to completely remove cell debris. All serum samples were aliquoted and immediately stored at −80 °C until investigation. Total RNA was isolated from each specimen using the miRNeasy Serum/Plasma Kit (Qiagen, Hilden, Germany) according to the manufacturer’s instructions.

#### 2.2.2. Profiling by miRNA Array Analysis

Reverse transcription (RT) was conducted using 2 μL RNA in a 10 μL reaction using miScript II RT kit (Qiagen, Hilden, Germany) according to the manufacturer’s instructions. The complementary DNA (cDNA) obtained from miRNA was amplified using the miScript miRNA PCR Array (MIHS-105Z, Qiagen, Hilden, Germany). Real-time polymerase chain reaction (RT-PCR) was performed on the StepOnePlus^TM^ Real Time PCR System (Applied Biosystems, Foster City, CA, USA) using the miScript SYBR Green PCR kit (Qiagen, Hilden, Germany). Thermal cycling conditions were 95 °C for 15 min, followed by 40 cycles of 95 °C for 15 s, 55 °C for 30 s, and 70 °C for 30 s. The data were analysed using PCR array data analysis tools (Qiagen, Hilden, Germany). These panels contained primers for the detection of the 84 most highly expressed miRNAs in human body fluids. Real-time PCR was performed according to the manufacturer’s instructions. In each 96-well plate, cDNAs from serum samples obtained during preoperative and postoperative month-6 time points from each patient were amplified in parallel. Expression values were calculated using the 2^−ΔCt^ method, and the mean and standard deviation of three technical replicates were presented. In addition, fold changes indicating values of changes in the expression of miRNAs between preoperative, postoperative 6-month, and postoperative 18-month time points were calculated. This fold change (relative expression) was analysed using the 2^−ΔΔCt^ method [20].

#### 2.2.3. Validation by Real-Time Quantitative Polymerase Chain Reaction (RT-qPCR)

RT of total RNA (miRNAs) was performed using miScript II RT kit according to the manufacturer’s instructions. Each 20 μL RT reaction contained 5 μL of extracted total RNA, 4  μL HiSpec buffer, 2  μL 10× nucleic mix, 2  μL of miScript RT enzyme. The reaction was incubated for 60  min at 37 °C and 5  min at 95 °C. RT-qPCR was performed on the StepOnePlus^TM^ Real Time PCR System using the miScript SYBR-Green PCR kit in a total reaction volume of 25 μL containing 2 μL cDNA (10× diluted). The miScript primer assays (Qiagen, Hilden, Germany) used in this stage are listed in Table 3. The PCR thermal parameters were as follows: A total of 15 min at 95 °C, 40 cycles of 15 s at 94 °C followed by 30 s at 55 °C, and 30 s at 70 °C. All qPCR reactions were run in triplicate for efficient test. To normalize the expression of the miRNAs being validated, the GeNorm algorithm (GenEx, Qiagen, Hilden, Germany) was used. *SNORD61* was used as an endogenous reference gene [21].

#### 2.2.4. Chemicals

The manufactured kits containing the chemicals required for each procedure according to the standard protocol provided by the manufacturer’s introductions. Reagents other than these kits were rarely used in the current study.

The chemicals and assay kits used for each procedure were summarized in Table 4.

#### 2.2.5. Data Analysis and Statistical Methods

Statistical analyses were performed using SPSS 23.0 software (IBM Corp., Armonk, NY, USA). Paired *t*-test for miRNA level analysis (^Δ^Ct) and a nominal *p*-value < 0.05 were used in the profiling stage, and Bonferroni corrected *p*-value (0.05/7) was used in the validation stage. To compare changes in postoperative levels of miRNAs associated with clinical and radiologic outcomes, a correlation analysis was performed using the Spearman rank test considering covariates, including age, gender, body mass index, preoperative K-L grade, and preoperative lower leg alignment. Wilcoxon signed-rank test was used for the comparison of clinical and radiological outcomes between preoperative and postoperative values.

## 3. Results

### 3.1. Identification of Differentially Expressed miRNAs in the Profiling Stage

A panel of 84 miRNAs and controls were profiled in the serum specimens obtained from six patients with medial compartment knee OA. The results of the expression of these 84 miRNAs based on biological groups are shown as a heat map (Figure 2). PCR array analysis revealed the expression of 34 of 84 target miRNAs in all serum specimens (Appendix A). Only three miRNAs, namely, miR-19b-3p, miR-29c-3p, and miR-424-5p, were upregulated with a significant fold change (Table 5). Hence, these were included in the validation stage.

### 3.2. Validation Stage: Serum Expression Levels of Seven miRNAs at Preoperative and Postoperative Month-6 Time Points in 27 Patients Who Underwent Open Wedge HTO

To validate our profiling results, the expression of seven miRNAs, the three miRNAs detected in the profiling stage and four other miRNAs (miR-140-3p, miR-454-3p, miR-let-7e-5p, and miR-885-5p) previously reported to be related to OA progression, was measured in the serum at preoperative and postoperative month-6 time points in 27 patients using RT-qPCR. Of them, four (miR-19b-3p, miR-140-3p, miR-454-3p, and miR-let-7e-5p) miRNAs showed differential expression patterns between 6 months after surgery and pre-operative time points following normalization with SNORD61 expression (Figure 3). We corrected the *p*-values using the Bonferroni method, and found that one miRNA (miR-19b-3p) from our profiling result and three miRNAs (miR-140-3p, miR-454-3p, and miR-let-7e-5p) from previous studies showed significant positive fold changes (Table 6). Raw data from all patients prior to HTO and 6 and 18 months after HTO are available in Appendix A.

### 3.3. Comparison of Clinical and Radiological Outcomes between Preoperative and Postoperative Month-6 Time Points in 27 Patients

Clinical outcomes assessed using a VAS and WOMAC questionnaire were significantly improved 6 months after surgery as compared with those reported before operation (Table 7). Preoperative pain, functional disability, and lower leg alignment showed significant improvements 6 months after index surgery in 27 patients.

### 3.4. Expression Changes in Seven miRNAs in the Serum 6 and 18 Months after Surgery in 20 Patients Who Were Recruited in the 18-Month Follow-Up Study

To further investigate the changes in the expression of these seven miRNAs up to 18 months after HTO, serum specimens were collected in 20 patients who were recruited and agreed to participate in an 18-month follow-up study. Their expression levels were measured using RT-qPCR. The expression of all miRNAs was upregulated after 18 months from surgery as compared with that detected at 6 months after surgery following normalization of values to SNORD61 expression. However, only two miRNAs (miR-424-5p and miR-140-3p) were significantly upregulated up to 18 months after surgery after correcting the *p*-values using the Bonferroni method (Table 8).

## 4. Discussion

HTO is widely implemented as a joint preservation surgery in medial compartmental OA and has been shown to improve articular cartilage after HTO in clinical studies [6,22]. Radiologic methods such as magnetic resonance imaging or second-look arthroscopic evaluation after HTO are used to assess cartilage repair process [23]. However, these evaluation techniques are expensive or invasive.

Generally, complex cytokine interaction is involved in arthritic cartilage damage. Proinflammatory cytokines (e.g., IL-1β, TNFα, IL-6, IL-15, IL-17, and IL-18) participate in the turnover of cartilaginous matrix, stimulating chondrocytes to increase the production of enzymes to enhance matrix degradation, mainly metalloproteinase, while IL-4, IL-10, and IL-13 inhibit this process. These cytokines are released by damaged cells of cartilage, bone tissue, and synovium and detected in the plasma and synovial fluid of patients. Genes related to the cartilage destruction in OA have been known to regulate the expression of cytokines [24]. Besides, the expression of these genes has been reported to be regulated by circulating miRNAs [8,9].

Recently, advanced techniques exploiting novel circulating miRNAs have highlighted the potential role of miRNAs as promising biomarkers, especially in relation to OA [8,9,10,11,25]. Cell-based or preclinical studies on OA have suggested that miRNAs were either diagnostic or underlying causative factors [26]. Circulating miRNAs have been detected in body fluids, and investigators have suggested their plausible effects on OA progression [27]. Non-invasive detection of circulating miRNAs specific to certain diseases may be of considerable help for disease diagnosis and prognosis. Therefore, we performed this study to identify serum miRNAs as potential prognostic biomarkers after HTO in medial compartmental OA and found significant changes in the expression patterns of certain miRNAs before and after surgery.

Four miRNAs, namely, miR-19b-3p, miR-140-3p, miR-454-3p, and miR-let-7e-5p, showed significant changes in their expression patterns in the validation study performed 6 months after HTO. Of these, only miR-140-3p was significantly upregulated up to 18 months after HTO. Therefore, miR-140-3p may be the most significant prognostic biomarker, although the other three miRNAs could be candidate prognostic biomarkers after HTO.

Both miR-9 and miR-140 families play important roles in normal cartilage development, and their aberrant expression is known to be involved in the pathology of OA [27]. We did not include miR-140 family in our inflammatory microarray panel, which is most commonly available in our country. Therefore, we added miR-140-3p in the validation phase. However, the expression changes in miR-9 and miR-140 may differ based on specimen type (tissue, SF, or serum).

Circulating miR-19b-3p has been previously implicated with OA severity [27]. The role of miR-19b-3p in inflammatory processes has been studied, and its clinical value as a potential biomarker for early diagnosis and prognosis of inflammatory diseases has been investigated [28]. Kong et al. reported the upregulation in the expression of miR-19b-3p in knee OA and the potential of the combined measurement of miR-19b-3p and other miRNAs for the diagnosis, and as a measure of severity of, knee OA [27]. Duan et al. recently found the underlying mechanism, wherein miR-19b-3p attenuates interleukin-1 beta (IL-1β)-induced extracellular matrix degradation and inflammatory injury in chondrocytes [29]. Consistent with previously published data about OA, miR-19b-3p has been related to clinical and radiological improvements after HTO in our study.

Both miR-29c-3p and miR-424-5p identified in the profiling phase of our study showed no significant fold changes in their expression patterns in the validation phase with 27 patients. Li et al. compared the expression of miRNAs in the SF between early- and late-stage knee OA and found significant changes in miR-29c-3p levels among many SF miRNAs [30]. However, our study aimed at mixed OA patients (early and late; K-L grade II and III) who underwent HTO and serum specimens. Therefore, we believe that miR-29c-3p is not a relevant serum biomarker after HTO, although it may be related to OA progression. Rousseau et al. evaluated the association between circulating miRNAs and knee OA in women and reported no significant changes in miR-29c-3p levels, consistent with our results [31]. Many studies have revealed the role of miR-424-5p in cell- or tissue-based OA [32]. miR-424-5p expression was downregulated in cartilage-derived stem cells from a degraded cartilage that led to reduced proliferation and changes in the differentiation profile of stem cells, consequently contributing to homeostasis imbalance and OA-related cartilage erosion. In our study, miR-424-5p expression was significantly upregulated 18 months after surgery as compared with that after 6 months from surgery. However, we believe that it is not a relevant prognostic biomarker because we failed to notice any significant fold change in its expression in the validation phase.

In addition to the above three identified miRNAs, four other previously known miRNAs were validated in 27 patients. The changes in clinical symptoms and radiological measurements were measured to verify any clinical improvement after HTO. From these seven miRNAs, four were identified as circulating miRNA signatures in the validation stage and could be related to clinical and radiological improvements after HTO in medial compartmental OA knee [30]. These have been known to be circulating miRNAs differentially expressed in OA. Among the seven miRNAs, one from the profiling stage (miR-19b-3p) and three added at the validation stage (miR-140-3p, miR-454-3p, and miR-let-7e-5p) showed significantly upregulated expression after 6 months from surgery as compared with that prior to surgery. Therefore, our results were consistent with those of previous OA studies that used serum specimens, especially for miR-140-3p [33]. As miR-140 has been known as a major miRNA implicated in OA and highly and selectively expressed in the cartilage, it has been the focus of many studies [34,35,36,37]. miR-140-3p is a circulating miRNA in the serum and related to OA. Target gene analysis of miR-140-3p revealed regulation of metabolic processes in OA pathology and that miR-140-3p expression was consistently downregulated in articular cartilage with OA severity [38]. miR-140-3p also plays a role in IL-1β-stimulated OA chondrocytes and shows inverse correlation between miR-140-3p and IL-1β-stimulated OA chondrocytes [39]. In our study, miR-140-3p expression was significantly upregulated up to 18 months after HTO, suggestive of its potential as a prognostic biomarker after HTO in medial compartmental knee OA. The level of miR-454-3p in the serum has been previously evaluated in OA, but no reports related to knee OA are yet published [40,41]. Although significant changes in circulating miR-454-3p level have been previously observed, such effects have been related to many other conditions, including tumour [42,43], inflammatory process [44,45], and nutrition [46]. Therefore, the role of miR-454 in prognosis after HTO in medial compartmental knee OA should be confirmed with additional studies.

Aside from the miR-140 family, miR-let-7e-5p has been suggested as an important circulating serum miRNA in OA [12]. In our study, miR-let-7e-5p expression was significantly upregulated and was consistent with the clinical and radiological improvements after HTO. This observation is in line with that of another study that identified miR-let-7e-5p as a negative predictor of OA [9]. Kung et al. suggested an increase in circulating miR-let-7e-5p level in patients with arthritis [12], while Beyer et al. revealed the downregulation in its expression in patients with OA [9]. Taken together, miR-let-7e-5p could also be a prognostic biomarker for OA improvement after HTO. However, previous reports are inconsistent, and our finding was based on the clinical setting of HTO patients. Accordingly, further studies are needed to eliminate any bias.

For clinical relevance, the possible correlation between clinical and radiological improvements and the expression of seven miRNAs was analysed in 27 patients. No direct correlation was noted between expression changes in serum miRNAs and outcomes, probably owing to the small number of patients and the subjective nature of clinical scoring.

Our study has several limitations. In the profiling phase, we used an 84 miRNA panel, which was available in our country, known to be primarily involved in inflammatory processes. Therefore, only 84 miRNAs were screened in this phase. As a compensation strategy, we included several important miRNAs reported to be related to OA progression in the validation phase. Future studies in a larger patient cohort are needed to confirm our findings, as only a small number of serum miRNAs have been analysed. Second, no positive control was included in our study that may be needed, as the change in each miRNA in the serum would be a result of the surgical procedure and not entirely specific to HTO. In this direction, we compared not only preoperative and postoperative expression levels of miRNAs but also clinical and radiologic improvements specific to HTO. Third, the underlying mechanism between miRNAs and post-HTO prognosis was not evaluated. We believe that our study provides a fundamental basis to identify potential prognostic miRNAs on the basis of changes in their expression levels along with clinical and radiological improvements after HTO in medial compartmental knee OA. Fourth, some miRNAs which have been known as important OA-related miRNAs may have been initially excluded because their expression may not have been significantly affected after HTO. HTO is performed at a stage where further OA progression is inevitable, and its effect is not to completely cure OA but to delay its progression through partial repair process of damaged cartilage via realignment of mechanical axis. Therefore, we believe that the change of miRNAs expression after HTO would be different from the physiologic change in progression of OA Fifth, the radiographic semiquantitative evaluation before and after HTO is a bit unprecise hence it is difficult to expect a direct correlation. It is thought that more precise results would have been obtained if MRI had been performed in all patients at final follow-up. Sixth, we could not completely exclude patients with mild OA or degenerative diseases in other joints or spine but no symptoms or remarkable changes on plain radiographs although we tried to rule out patients with degenerative diseases in other joints through the inclusion criteria. Therefore, mild degenerative diseases undetected among enrolled patients may have affected our results. Finally, this study focused on serum miRNAs as a measure of improvement in OA after HTO. The measurement of the expression of miRNAs involved in cartilage repair and degradation may be more accurate and reliable in SF or cartilage tissue than in serum. However, to obtain SF or cartilage tissue specimen is more invasive, difficult, and complicated as against serum specimens that can be easily and more consistently obtained. Accordingly, we believe that the identification of serum miRNAs related to cartilage repair as prognostic biomarkers is of paramount significance.

miRNAs in body fluids have been used as potential biomarkers to evaluate OA severity and progression in addition to clinical and radiological assessments. Our study is the first to screen circulating miRNAs in the serum from patients with medial compartment knee OA and identify a panel of serum miRNAs that can help evaluate improvements in OA between before and after joint preservation surgery. Therefore, the present study suggests the potential use of these serum miRNAs as prognostic biomarkers to predict clinical improvement in cartilage repair after HTO in medial compartmental knee OA. However, further studies are warranted to establish clinical implications of changes in the expression of these miRNAs after HTO.

## 5. Conclusions

Four serum miRNAs—miR-19b-3p, miR-140-3p, miR-454-3p, and miR-let-7e-5p—were significantly upregulated in the validation phase of our study and identified as potential prognostic biomarkers in medial compartmental knee OA patients showing clinical and radiological improvements after HTO. Of these, miR-140-3p level was significantly upregulated up to 18 months after HTO, indicating that miR-140-3p may be the best prognostic biomarker after HTO in medial compartmental knee OA. We believe that our results provide a novel perspective on circulating serum miRNAs to predict cartilage repair along with clinical and radiological improvements after HTO.

## Figures and Tables

**Figure 1 diagnostics-11-00258-f001:**
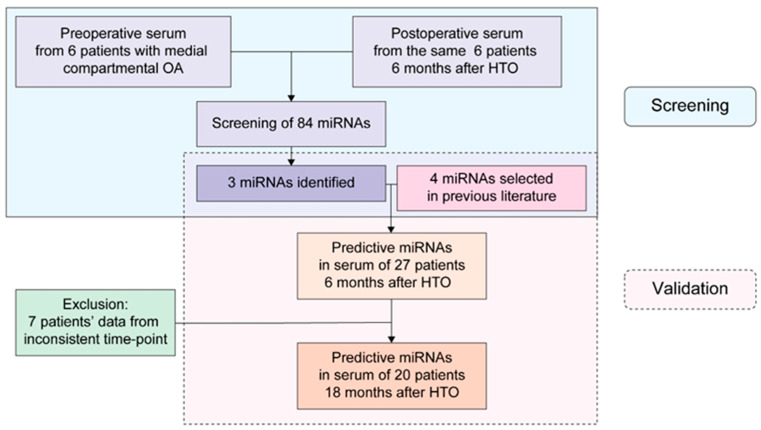
Study design. Three candidate miRNAs were identified in the serum from six patients with high tibial osteotomy at preoperative and postoperative time points. We evaluated their expression together with other four previously identified miRNAs (total seven miRNAs) in 27 patients at postoperative 6 months and 20 patients at postoperative 18 months for validation. Data of seven patients were excluded, owing to different collection time points.

**Figure 2 diagnostics-11-00258-f002:**
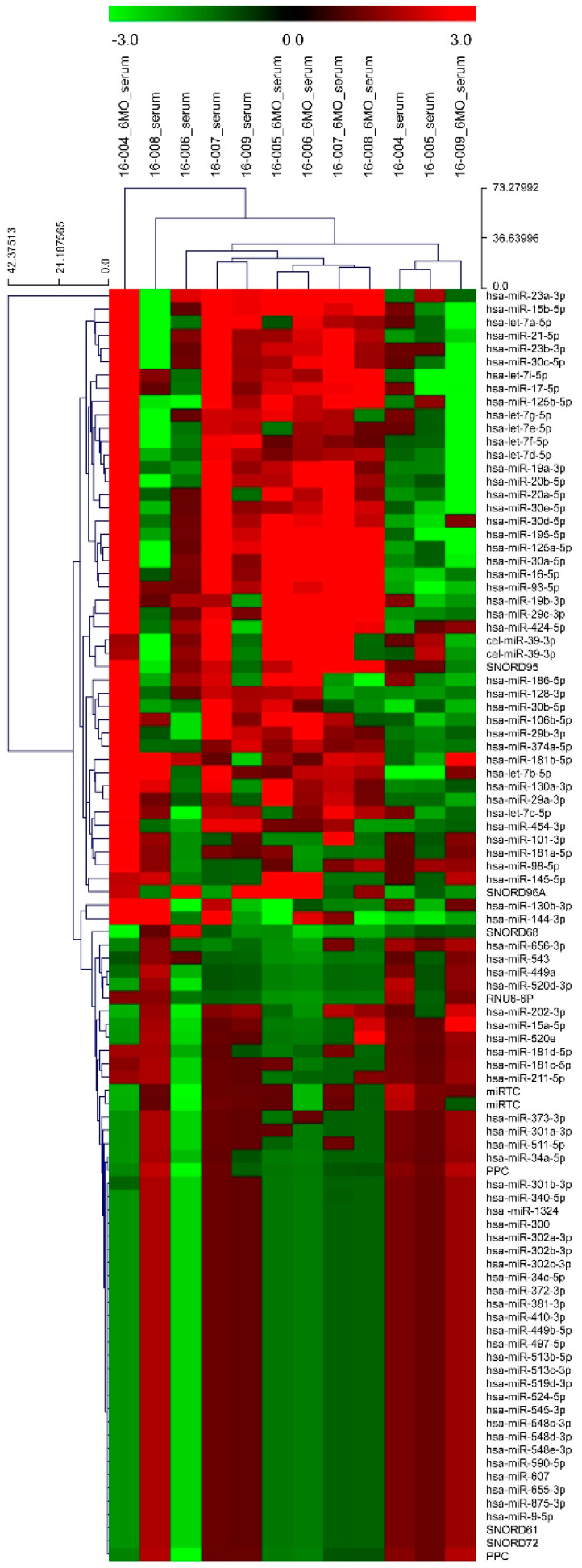
Heatmap showing the expression levels of 84 target miRNAs from miRNA PCR array screening performed for serum samples collected from six patients at preoperative and postoperative month-6 time points.

**Figure 3 diagnostics-11-00258-f003:**
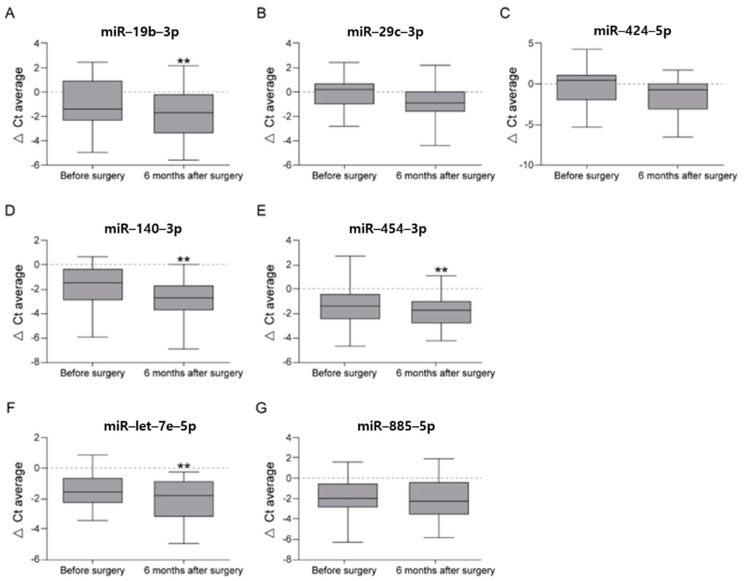
Box plot showing the interquartile range and median values of the expression levels of miR-19b-3p (**A**), miR-29c-3p (**B**), miR-424-5p (**C**), miR-140-3p (**D**), miR-454-3p (**E**), miR-let-7e-5p (**F**), and miR-885-5p (**G**) before surgery and 6 months after surgery at the validation stage. The values represent the relative levels normalized to the level of the reference gene, *SNORD61*. ** indicates a Bonferroni-corrected *p*-value < 0.0071 (0.05/7) corresponding to the statistical significance.

**Table 1 diagnostics-11-00258-t001:** Data of six patients at profiling stage.

Serial Number	Gender	Age (Year)	Body Mass Index (kg/m^2^)	Kellgren–Lawrence Grade	Visual Analogue Scale	WOMAC Score			
						Total	Pain	Stiffness	Function
1	F	57	28.4	II	6	54	13	4	37
2	F	55	23.9	II	5	10	1	0	9
3	F	61	22.8	III	7	56	8	6	42
4	F	56	27.9	III	7	42	9	3	30
5	F	65	30.2	III	3	14	0	0	14
6	F	52	26.0	III	4	33	9	3	21

WOMAC: Western Ontario and McMaster Universities Osteoarthritis Index.

**Table 2 diagnostics-11-00258-t002:** Demographic data of 27 patients 6 months after HTO and 20 patients 18 months after HTO at validation stage.

Serial Number (POD 6 mo)	Serial Number (POD 18 mo)	Gender	Age (Year)	Body Mass Index (kg/m^2^)	Kellgren–Lawrence Grade	Visual Analogue Scale	WOMAC Score			
							Total	Pain	Stiffness	Function
1		F	57	23.4	III	6	14	3	2	9
2	1	F	58	30.1	II	3	6	2	0	4
3	2	F	50	30.5	III	3	21	2	2	17
4	3	F	46	33.3	III	7	22	3	2	17
5		F	54	26.5	II	7	26	5	2	19
6	4	F	51	25.5	II	9	59	14	3	42
7		F	56	21.8	III	8	28	5	5	18
8		F	57	28.4	II	6	54	13	4	37
9	5	F	55	23.9	II	5	10	1	0	9
10	6	F	61	22.8	III	7	56	8	6	42
11	7	F	56	27.9	III	7	42	9	3	30
12		F	52	26	III	4	33	9	3	21
13	8	M	64	30.9	III	7	30	7	2	21
14		F	55	24	III	8	48	10	4	34
15	9	F	56	26.4	III	7	52	9	6	37
16	10	M	61	28.5	III	8	53	12	3	38
17	11	F	59	26.3	III	5	36	8	3	25
18	12	F	56	27.5	III	5	42	15	6	21
19	13	F	61	29.7	III	8	61	13	6	42
20	14	F	57	31.2	III	6	45	10	4	31
21		M	58	25.4	III	5	23	7	3	13
22	15	F	56	21.8	III	3	29	5	2	22
23	16	F	56	25.4	III	2	26	5	2	19
24	17	M	59	24.8	III	8	37	7	4	26
25	18	F	60	25.2	III	7	37	2	4	41
26	19	M	64	25.7	III	7	42	5	0	37
27	20	M	56	25.4	III	4	25	4	3	18

HTO: high tibial osteotomy.

**Table 3 diagnostics-11-00258-t003:** Specific miScript primer assays used for RT-qPCR.

miRNA Symbol	Assay ID	Catalogue No.	Sequencing
SNORD61	Hs_SNORD61_11	MS00033705	
miR-19b-3p	Hs_miR-19b_2	MS00031584	5′UGUGCAAAUCCAUGCAAAACUGA
miR-29c-3p	Hs_miR-29c_1	MS00003269	5′UAGCACCAUUUGAAAUCGGUUA
miR-424-5p	Hs_miR-424_1	MS00004186	5′CAGCAGCAAUUCAUGUUUUGAA
miR-140-3p	Hs_miR-140-3p_1	MS00008673	5′UACCACAGGGUAGAACCACGG
miR-454-3p	Hs_miR-454_1	MS00007861	5′UAGUGCAAUAUUGCUUAUAGGGU
miR-let-7e-5p	Hs_let-7e_3	MS00031227	5′UGAGGUAGGAGGUUGUAUAGUU
miR-885-5p	Hs_miR-885-5p_1	MS00010668	5′UCCAUUACACUACCCUGCCUCU

**Table 4 diagnostics-11-00258-t004:** Chemicals or assay kits used in this study.

Procedure	Chemical/Assay Kit
RNA preparation	miRNeasy Serum/Plasma kit (Qiagen)
	miRNeasy Serum/Plasma Spike-In Control (Qiagen)
	QIAzol Lysis Reagent (Qiagen)
	Absolute Ethanol (Merck)
	Chloroform (without added isoamyl alcohol) (Sigma)
PCR array	miScript miRNA PCR array (MIHS-105Z) (Qiagen)
Reverse Transcription	miScript II RT kit (Qiagen)
Real-Time PCR	miScript SYBR Green PCR kit (Qiagen) Specific primer assays (Table 3)

**Table 5 diagnostics-11-00258-t005:** 3 miRNAs showing significant fold change at profiling stage.

miRNA	Preop.		Postop. Month 6		Fold Regulation	*p*-Value *
	ΔCt	2^−ΔCt^	ΔCt	2^−ΔCt^		
hsa-miR-19b-3p	−0.66 ± 0.87	1.57	−2.47 ± 1.38	5.53	3.51	0.011
hsa-miR-29c-3p	−0.32 ± 0.79	1.25	−2.10 ± 1.26	4.28	3.43	0.015
hsa-miR-424-5p	−0.48 ± 1.17	1.39	−2.23 ± 0.80	4.69	3.37	0.021

Values are presented as the mean ± standard deviation. * *p* < 0.05.

**Table 6 diagnostics-11-00258-t006:** Fold changes in the expression of seven miRNAs at profiling and validation stages. Data from the validation stage were collected 6 months after surgery.

miRNA	Profiling Stage Using Microarray	*p*-Values *	Validation Stage Using Real-Time PCR	*p*-Values *
	Fold Change		Fold Change	
has-miR-19b-3p	3.51	0.011	3.66	0.003 **
has-miR-29c-3p	3.43	0.015	3.68	0.01
has-miR-424-5p	3.37	0.021	5.49	0.063
has-miR-140-3p	-	-	4.12	0.005 **
has-miR-454-3p	-	-	2.18	0.004 **
has-miR-let-7e-5p	-	-	2.38	0.004 **
has-miR-885-5p	-	-	2.06	0.096

* Paired *t*-test. ** indicates a Bonferroni-corrected *p*-value < 0.0071 (0.05/7) corresponding to the statistical significance.

**Table 7 diagnostics-11-00258-t007:** Improvements in radiological and functional outcomes in 27 patients at 6 months after surgery.

Variables	Preoperative	Postoperative Month-6	*p*-Value *
Mechanical axis (−: varus, +: valgus)	−5.0 ± 2.2	+2.5 ± 1.5	<0.001
Weight-bearing line ratio (%)	18.6 ± 6.2	59.5 ± 8.2	<0.001
Visual analogue scale	6.0 ± 1.9	2.6 ± 1.1	<0.001
WOMAC total	36.4 ± 16.0	24.2 ± 11.1	0.002
Pain	7.8 ± 4.0	4.3 ± 2.5	0.001
Stiffness	3.2 ± 1.7	2.2 ± 1.3	0.022
Function	25.4 ± 11.4	17.8 ± 8.5	0.006

* *p*-value < 0.05.

**Table 8 diagnostics-11-00258-t008:** Comparison of the expression of miRNAs using RT-PCR at post-operative months 6 and 18.

miRNA	6-Month	18-Month	*p*-Value *
	Fold Change	Fold Change	
has-miR-19b-3p	2.99	5.91	0.012
has-miR-29c-3p	2.3	4.44	0.016
has-miR-424-5p	5.13	8.6	0.003 *
has-miR-140-3p	3.09	8.48	0.003 *
has-miR-454-3p	2.6	3.57	0.027
has-miR-let-7e-5p	1.25	2.38	0.071
has-miR-885-5p	1.31	2.93	0.153

* Wilcoxon signed rank test. A Bonferroni corrected *p*-value < 0.0071 (0.05/7), which is marked with an asterisk, was considered statistically significant.

## Data Availability

The data presented in this study are available on request from the corresponding author.

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
