# Peer review of "Significant Changes in Serum MicroRNAs after High Tibial Osteotomy in Medial Compartmental Knee Osteoarthritis: Potential Prognostic Biomarkers"

_diagnostics, 2021, doi:10.3390/diagnostics11020258_

Round 1

Reviewer 1 Report

The Study showed an interesting differential expression of serum miRNAs investigation before and after high tibial osteotomy to identify potential miRNAs as prognostic biomarkers. There are few minor comments need to be addressed:

Page 1; Line 26: Abstract section: Significant figures should be given either in the percentage or statistically significant values.

Page 2; Line 77: Insert a separate section as "Chemicals" which should include all the chemicals, assay kits used in the study.

Reviewer 2 Report

This paper focuses on evaluating if selected miRNAs could be used as candidates for OA progression/development/improvement following HTO. The authors identified four candidate miRNAs which could be potentially used to evaluate if a given medical intervention is going to be beneficial in the context of subsequent OA progression.

The study is well-designed, the methodology is well described, yet the paper needs some improvements in the materials section adn the discussion. I have three main problems with the text

  1. The inclusion criteria are not well described. There is a large number of factors potentially affecting miRNA levels (ex. OA of the pat-fem joint in your patients, OA of other joints). I am sure the authors did verify them but failed to precisely describe it in the study (see below for more details). I am sure this can be fixed by adding a few sentences.
  2. The authors write about cartilage regeneration in several paragraphs. As a humble orthopaedic surgeon I do respect their enthusiasm for regenerative medicine, yet having performed quite a few HTOs/UNIs – and TKAs following these procedures I would advise the authors to tone down these statements. Cartilage does have some limited regenerative potential (as far as we know by now) yet of extremely limited clinical significance in OA – as most authors would agree. Consequently the “regeneration” of cartilage following HTO is – to say the least – controversial ( for more details see below) and, as I believe, should be described in a more neutral fashion.
  3. The last and least concerning point - overall this paper seems to give the impression that OA is a wear-and-tear phenomenon (just like polyethylene wear in a hip replacement) which can be qualified by miRNA level examination. I am fully aware that the authors know this better than I do, and out of the respect for the time of the readers did not write about the complex cytokine interplay in OA. Still – I would advise them to add a few sentences that performing a HTO/Uni or an other procedure which “relieves” the patient of localized knee OA cannot – at least in some cases – prevent OA development in other compartments, consequently the findings in this study may not apply to all “real life scenarios”, and that there is most likely no such thing as a “universal OA marker” because of the complexity of the cytokine network.

A few minor points/details

37 – please rewrite  -  leading to its permanent damage

41 – please tone down this and subsequent parts regarding cartilage regeneration; most authors would strongly disagree your statements. Procedures such as HTO can result in decreased mechanical loading to a given part of cartilage, yet there is no evidence that this will result in growth of new layers of cartilage , as is suggested by the term regeneration (ex. as would be expected for regeneration of bony tissue). Please consider using terms such as “repair process”, ”regenerative process” instead

Line 99 – please explain the abbreviation “MA” – I admit that it is explained to some degree further down the page but it would be easier to read the text with the full name given here.

Section 2.1 – Please provide inclusion criteria for HTO. What about other knee injuries – ACL/menisci – perhaps they did play a role in the inflammatory process in the knees and could have influenced your results.

Line 85 – did you use conventional/long standing/Rosenberg X-rays ?

Section 2.2.1 – Please clarify -  did you use any RNA stabilizing agents (such as RNAlater or similar) ?

Section 3.1 – the calibration group – please clarify if the selected microRNA from your group there was a significant fold change in the selected miRNAs

  • please consider modifying table 3 - my suggestion would be to leave the three selected miRNAs here (similarly as is done in table 4), and removing the rest of data. Publishing the removed part as supplementary material would really improve the papers overall clarity

Discussion

  • Please add 1-2 sentences regarding the complex nature cytokine interaction in arthritic cartilage damage. Your paper gives the impression that this is almost a simple wear-and-tear phenomenon and its activity can be assessed using miRNA. Consider this recent open-access paper from MDPI as areference doi: 10.3390/ijms21155430.
  • Please expand a bit the limitation described in lines 335-344, as I humbly believe be discussed - It is generally believed that at some point OA is a one-way street – at a certain “joint damage threshold” it will be progressing regardless of any medical interventions. Perhaps some of the patients from your study reached this point ? My point is – perhaps some of the initially rejected miRNAs you examined are indeed markers of OA (confirm the metabolic shift from homeostasis to arthritis), yet performing HTO at a stage where further OA progression is inevitable will not affect their levels significantly, consequently the rejects should still be considered by other authors as possible candidates for OA markers, while – as you wrote in lines 310-320 you focused on those who are markers of improvement
  • Please add some minior limitations – a) the radiographic semiquantitative evaluation is (as a diplomat would say) a bit unprecise hence it is difficult to expect a direct correlation, perhaps including MRI would yield more precise results; b) the inclusion criteria are imprecise – perhaps some/all patients had anterior compartment OA – this would not be affected by the HTO and consequently a less pronounced miRNA profile shift would be seen. Perhaps some patients had ACL deficiency – and instability would result in shear forces being applied to cartilage ant potentially affecting miRNA levels ? Please address these issues c) were your patients screened for OA in other joints – hip ? spinal degenerative disc disease with facet joint OA ? Could this have an effect of the results ?

Round 2

Reviewer 2 Report

Thank you for sending the revised version of ther paper. Generally it was substantially improved, I really enjoyed reading it.